# Surveillance Study of Hepatitis E Virus (HEV) in Domestic and Wild Ruminants in Northwestern Italy

**DOI:** 10.3390/ani10122351

**Published:** 2020-12-09

**Authors:** Andrea Palombieri, Serena Robetto, Federica Di Profio, Vittorio Sarchese, Paola Fruci, Maria Cristina Bona, Giuseppe Ru, Riccardo Orusa, Fulvio Marsilio, Vito Martella, Barbara Di Martino

**Affiliations:** 1Faculty of Veterinary Medicine, Università degli Studi di Teramo, 64100 Teramo TE, Italy; apalombieri@unite.it (A.P.); fdiprofio@unite.it (F.D.P.); vsarchese@unite.it (V.S.); pfruci@unite.it (P.F.); fmarsilio@unite.it (F.M.); 2Istituto Zooprofilattico Sperimentale del Piemonte, Liguria e Valle d’Aosta (IZS PLV), 11020 Quart AO, Italy; serena.robetto@izsto.it (S.R.); cristina.bona@izsto.it (M.C.B.); giuseppe.ru@izsto.it (G.R.); riccardo.orusa@izsto.it (R.O.); 3Department of Veterinary Medicine, Università Aldo Moro di Bari, Valenzano, 70121 Bari BA, Italy; vito.martella@uniba.it

**Keywords:** hepatitis E virus (HEV), domestic and wild ruminants, HEV antibodies, viral RNA

## Abstract

**Simple Summary:**

Hepatitis E virus (HEV) infection can cause both acute and chronic hepatitis in humans and represents an emerging public health concern worldwide. In developed countries, zoonotic transmission of HEV genotypes 3 and 4 is caused by ingestion of raw or undercooked meat of infected swine or wild boars, the main reservoirs of HEV. However, in the last few years, molecular and serological evidence seem to indicate that several other animal species may act as HEV host, including domestic and wild ruminants. In this study, serum and fecal specimens from sheep, goats, red deer, roe deer, chamois, and Alpine ibex collected in two northwestern Italian regions (Piemonte and Valle d’Aosta) were screened molecularly and serologically. With the exception of chamois, HEV antibodies were found both in the domestic and wild ruminant species investigated with the highest rates in sheep and goats. These findings demonstrate that wild also domestic ruminants may be implicated in the viral cycle transmission.

**Abstract:**

In industrialized countries, increasing autochthonous infections of hepatitis E virus (HEV) are caused by zoonotic transmission of genotypes (Gts) 3 and 4, mainly through consumption of contaminated raw or undercooked pork meat. Although swine and wild boar are recognized as the main reservoir for Gt3 and Gt4, accumulating evidence indicates that other animal species, including domestic and wild ruminants, may harbor HEV. Herein, we screened molecularly and serologically serum and fecal samples from two domestic and four wild ruminant species collected in Valle d’Aosta and Piemonte regions (northwestern Italy. HEV antibodies were found in sheep (21.6%), goats (11.4%), red deer (2.6%), roe deer (3.1%), and in Alpine ibex (6.3%). Molecular screening was performed using different primer sets targeting highly conserved regions of hepeviruses and HEV RNA, although at low viral loads, was detected in four fecal specimens (3.0%, 4/134) collected from two HEV seropositive sheep herds. Taken together, the data obtained document the circulation of HEV in the geographical area assessed both in wild and domestic ruminants, but with the highest seroprevalence in sheep and goats. Consistently with results from other studies conducted in southern Italy, circulation of HEV among small domestic ruminants seems to occur more frequently than expected.

## 1. Introduction

Hepatitis E virus (HEV) infection is a major health problem worldwide. HEV typically causes self-limiting acute viral hepatitis, although chronic infection with neurological and other extrahepatic manifestations has also been reported [1]. HEV is a quasi-enveloped [2,3] single-stranded positive-sense RNA virus of approximately 27 to 34 nm in diameter, classified in the family *Hepeviridae*, genus *Orthohepevirus*, which comprises 4 species, *Orthohepevirus A* to *D* [4]. Based on the full-length genome analysis, HEV strains within the species *Orthohepevirus A* have been assigned to at least eight distinct genotypes (Gt1–Gt8) [5], with four major Gts (1–4) implicated in human infection. Gt1 and Gt2 are restricted to humans and associated with large, waterborne outbreaks of disease in tropical and subtropical areas [1]. In contrast, Gt3 and Gt4 are zoonotic and cause sporadic and cluster cases of hepatitis E in both industrialized and developing countries [6,7]. Gt5 and Gt6 have been detected only in wild boars in Japan [8], whilst Gt7 and Gt8 from dromedary camels in United Arab Emirates [9] and from Bactrian camels in China [10], respectively. Except for Gt7, identified from a chronically infected human liver transplant patient who consumed camel milk and meat [11], the zoonotic potential of Gt5, Gt6, and Gt8 is still unclear.

Consumption of poorly cooked or raw pork meat is considered the major source of human infection by Gt3 and Gt4 HEVs with domestic pigs and wild boars identified as the main animal reservoirs [12]. Since the first identification of Gt3 HEV in swine [13], several molecular and serological surveys showed high prevalence in pigs and wild boars worldwide [12]. In Europe, investigations performed in swine herds revealed seroprevalences estimated between 30% and 100% [14,15,16,17,18,19,20,21,22,23] with molecular detection rates of 0.9% to 87.5% [24,25,26,27,28,29,30,31,32,33]. Similarly, epidemiological studies performed in wild boar populations reported antibody detection rates ranging from 12.5% to 57.4% and molecular prevalence of 0.3% to 68.2% [12,19,34,35,36,37,38]. Transmission from deer to humans has also been described [39,40], although they mostly undergo spill-over HEV infections in contaminated habitat shared with wildlife reservoirs [12,41]. Evidence for HEV zoonotic transmission by ingestion of uncooked deer meet was first reported in 2003 in Japan [39] during an outbreak of acute hepatitis involving four members of the same family that consumed deer raw meat (Sika deer, *Cervus nippon nippon*). Molecular analysis of frozen leftover meat portions revealed the presence of Gt3 HEV RNA, showing 100% nucleotide identity to the sequences identified in human patients [39].

Molecular surveys performed in wild ruminants in Europe have reported prevalence ranging from 1.2% to 16.0% in red deer (*Cervus elaphus*) [41,42,43,44,45,46,47,48,49] and from 3.3% to 34.4% in roe deer (*Capreolus capreolus*) [41,42,43,49,50]. Accumulating literature worldwide suggests that besides wilds, domestic ruminants, like sheep and goats, may also harbor HEV [51,52,53,54,55,56,57], posing additional risks for zoonotic infection. Gt4 HEV sequences genetically closest to strains identified in swine and humans, were first detected in liver specimens collected from slaughtered sheep in the Xinjiang region (China) in a 2015 study [56]. Subsequently, similar strains have been also found in stool, serum and milk samples collected from goat herds in Yunnan Province (China) [52]. In the European continent, the identification of Gt3 strains in goat farms has been already documented in Italy [51], during a survey performed in a restricted geographical area (Abruzzo, southern Italy). In the same area, HEV has also been detected at high rate in wild boars [58]. Serological and molecular screening of seven small sheep farms located in the same region has revealed seroprevalence rates ranging from 6.6% to 38.3% and has identified HEV strains genetically related to those detected locally in goats, wild boars, and humans [54,59].

In this study, in order to gather data on HEV epidemiology in other geographical areas, a large serological and molecular investigation was performed by assessing collections of serum and fecal samples from domestic and wild ruminants sharing the same habitats in two northwestern Italian regions (Valle d’Aosta and Piemonte).

## 2. Materials and Methods

### 2.1. Sampling

A total of 416 domestic and wild ruminants were screened for the presence of HEV antibodies and viral RNA. In more detail, between September 2017 and December 2019, serum and fecal samples were collected from 134 sheep and 167 goats, collected respectively in five ovine and sixteen caprine farms from 5 municipalities in the area of Cuneo (Piemonte region) and from 15 municipalities in the Valle d’Aosta region (Figure 1a). The flock size ranged from 6 to 75 sheep and from 5 to 90 goats. All animals, free to graze on pastures, were clinically healthy at the time of sampling and were divided on the basis of age in groups <3 years, 3–4 years, and >4 years. The circulation of HEV in wild ruminants was investigated mainly in the Valle d’Aosta region, where ungulates wildlife are very abundant and contact between them and domestic livestock can occur easily since pastures overlap with the feeding areas of red deer (*Cervus elaphus*), roe deer (*Capreolus capreolus*), and chamois (*Rupicapra rupicapra*). Briefly, one hundred and fifteen paired fecal and serum specimens were collected from 38 red deer, 32 roe deer, and 13 chamois sampled during the regular hunting season (from September 9 to December 18) in Valle d’Aosta and submitted to the National Reference Centre for Wild Animal Diseases (CeRMAS—IZS PLV). In addition, 32 Alpine ibex (*Capra ibex*) samples obtained during species control activity in province of Torino (Piemonte Region) (*n* = 2) and in Valle d’Aosta (*n* = 30), were included in the serological and molecular screening (Figure 1b). Fecal and serum specimens were placed in isothermal boxes using ice bags, transferred in the lab and kept frozen at −80 °C until tested.

### 2.2. Serological Assay

All serum samples were screened for the presence of HEV antibodies by using a species-independent commercial double-antigen sandwich enzyme-linked immunosorbent assay (ELISA) kit (Wantai Biological, Beijing, China), based on the peptide E2 spanning amino acids 394–606 of the major structural protein of HEV Gt1 (GenBank accession number D11092) [60]. ELISA test was performed following all the manufacturer’s instructions. Absorbance was measured at 450 nm (OD_450_) using a Multiskan automatic plate reader (ThermoLabsystems, Abu Gosh, Israel). For each test, the cut-off value was calculated as Nc + 0.12 (Nc = the mean absorbance value for three negative controls).

When comparing with other HEV antibody detection ELISAs, the assay employed in this study was found to have a high sensitivity, although it seems to possess a lower specificity [61].This antigen, widely used to detect Gt1–Gt4 HEV antibodies in humans, has also been successfully used to evaluate HEV seroconversion in animals infected either under natural or experimental conditions with novel members of the species Orthohepevirus A, such as Gt7 and Gt8 [62,63], as well as to detect specific IgG antibodies in ferrets and foxes that tested positive, on molecular investigations, for Orthohepevirus C RNA [64,65].

### 2.3. Molecular Assay

Fecal samples were homogenized in phosphate-buffered saline (0.15 M, pH 7.2) to obtain a total of 1 mL and then centrifuged at 10,000× *g* for 3 min. Each fecal specimen was seeded with 50 μL of feline calicivirus strain F9 (ATCC^®^ VR-782™) at the final titer of 4 × 10^6^ as RNA extraction control, and to test for the presence of inhibitors. Total RNA was extracted individually from each fecal (0.5 mL) and serum (0.5 mL) specimen by using the TRIzol LS (Invitrogen, Ltd., Paisley, UK), following the manufacturer’s instructions.

The presence of *Orthohepevirus A* RNA was assessed by a quantitative reverse transcription PCR (qRT-PCR), targeting a conserved 68 nucleotide region of ORF3 gene, as previously described [66]. Viral RNA quantification was performed using the TaqMan Fast Virus 1-Step Master Mix (Invitrogen Ltd., Milan, Italy) in a 25-μL volume comprising 5 μL of extracted RNA and 20 μL of master mix. Primers (JVHEVF: 5′-GGTGGTTTCTGGGGTGAC-3′ and JHEVR: AGGGGTTGGTTGGATGAA-3′) and TaqMan probe (JVHEVP: 5′-TGATTCTCAGCCCTTCGC-3′) were used at concentrations of 200 and 100 nM, respectively. In order to standardize the system, the WHO international standard for HEV RNA (code 6329/10) was used. Tenfold serial dilutions (from 10^0^ to 10^8^ copies) of a plasmid containing the 68 bp ORF3 fragment of a Gt3 HEV wild boar strain (HEV/WB/P6-15/ITA, accession no. KU508285) [58] were used in each PCR run.

All the samples were also tested by nested RT-PCR using the primer sets Fw1679/Rw1680 and intFw1681/intRw1682 [67] to amplify a 172-bp fragment of the methyltransferase region (ORF1) highly conserved within the species *Orthohepevirus A*.

Screening for other members of the genus *Orthohepevirus* was done by heminested RT-PCR (pan-hepevirus RT-PCR) using broadly reactive primers designed to amplify all members of the family *Hepeviridae* [68] and targeting a region of 338-bp of the viral RNA-dependent RNA polymerase (RdRp) complex.

### 2.4. Statistical Analysis

The data were analyzed using the Stata v.16.1 Software (StataCorp, 2016, www.stata.com). Calculation of the confidence intervals (CI) for the seroprevalence estimates was performed by Wilson score interval of 95%. Univariate and multivariate logistic regression analyses were carried out to determine the association between detection of HEV antibodies in sheep and goats and different age groups. A *p* value of <0.05 was considered statistically significant.

### 2.5. Ethics Statement

For this study, no ethical approval was required (Decreto Legislativo 4 March 2014, N. 26). Serum and fecal samples from wild ruminants were collected during the routinely monitoring activities performed by the National Reference Centre for Wild Animal Diseases (CeRMAS—IZS PLV), whilst samples from sheep and goats were collected only for veterinary diagnostic purposes.

## 3. Results

By screening 134 sheep serum samples, HEV antibodies were detected in a total of 29 sera with an overall prevalence of 21.6% (95% CI 15.0–29.6%) and OD_450_ values ranging from 0.17 to 1.96 (mean OD_450_ 0.49). When assessing goat sera, the positivity rate was 11.4% (95% CI 7.0–17.2%; 19/167) with OD_450_ comprised between 0.20 and 3.24 (mean OD_450_ 0.88). Analyzing the distribution of the seropositivity across the farms from the two regions investigated (Table 1), antibodies to HEV were found in 2/5 sheep herds, located in the Piemonte region, with rates respectively of 18.7% (95% CI 10.6–29.3%) and 36.6% (95% CI 22.1–53.1%), whilst they were not detected in 3 sheep farms from Valle d’Aosta. By converse, the prevalence rate in goats was 11.1% (10/90) in Piemonte and 11.7% (9/77) in Valle d’Aosta (Figure 2).

Information about age, available for a total of 286 animals, was used to estimate age-related difference in prevalence in both the species investigated (Figure 3). The highest rates were observed in the 3–4 years age group with rates of 13.6% for goats and 35.4% for sheep, whilst the lowest were detected in animals aged <3 years (11.9% and 13.1%, respectively). However, no statistically significant differences were found when compared the 3–4 years age group to the <3 years or >4 years age groups.

Among the four species of wild ruminants, HEV antibodies were found in red deer, roe deer, and Alpine ibex with prevalence of 2.6% (1/38), 3.1% (1/32), and 6.3% (2/32), respectively, and with OD_450_ values ranging from 0.19 to 1.0, while none of the sera collected from chamois reacted with HEV antigen (Table 2). All of the four positive sera were collected from animals sampled in the Valle d’Aosta region (Figure 4).

In order to explore the genetic heterogeneity of the HEVs circulating in the investigated geographical settings, all serum and fecal samples were screened molecularly. Out of the six ruminant species analyzed, HEV RNA was detected by qRT-PCR in four sheep fecal specimens (3.0%; 4/134) collected from the two herds in which HEV antibody detection rates were 18.7% and 36.6%, respectively. Viral loads ranging from 11 × 10^0^ to 9.8 × 10^1^ RNA copies/gram faeces. By re-screening fecal and serum samples by qualitative RT-PCR, HEV RNA was not detected in any of the tested animals either by using a nested RT-PCR for the species *Orthohepevirus A* or using a pan-hepevirus RT-PCR.

## 4. Discussion

This surveillance study was initiated to draw a more complete picture of HEV epidemiology in ruminants by monitoring geographical areas in which the ecological interaction between wildlife and extensive farming livestock may allow ideal conditions for circulation of pathogens.

On serological screening, HEV-specific antibodies were detected in five of the six ruminant species assessed in this study, suggesting that these species are exposed to natural infection with HEV throughout their life, even if with marked differences in seroprevalence between domestic and wild animals.

With the exception of chamois, HEV antibodies were detected in all the wild species, including a population of Alpine ibex, whose susceptibility to HEV infection had never been documented thus far. These findings, while extending the HEV host spectrum to an additional distinct mammalian species, raises questions about the the exact nature of this serological positivity, as only antibodies have been detected. Indeed, in our analysis, HEV molecular evidence was not obtained in any of the wild ruminants included in the screening. Furthermore, we found an overall low seroprevalence rate for HEV (3.5%; 4/115), confirming previous data collected during 2013–2015 surveillance from three distinct Italian Alpine regions [69], in which HEV antibodies were detected with rates of 1.2% (2/172) in chamois and 0.8% (2/254) in red deer. Accordingly, it could be hypothesized a limited role of these wild animals in maintaining HEV circulation, at least in the investigated geographical settings. This restricted viral circulation could account for the low HEV detection rates observed in molecular (3.7%; 12/320) and serological (4.9%; 29/594) investigations reported previously [70] by testing liver and serum samples collected from wild boar populations in northwestern Italy (Piemonte, Liguria, and Valle d’Aosta). In addition to foodborne transmission, several studies [71,72,73,74,75] suggest that contact exposure with infected animals may represent a possible source of infection. The exposure risk to HEV infection in hunters was documented in a study performed in central Italy [72]. Viral RNA was found in 33.5% of wild boar liver samples tested and HEV antibodies have been detected in sera from hunters with a prevalence rate of 25% (5/20), with three positive samples containing also HEV RNA. The general low prevalence detected in wild ungulates may imply that the zoonotic transmission risks for hunters in the area assessed in this study are limited.

A diverse scenario on HEV circulation was depicted in the population of small domestic ruminants investigated. Seroprevalence rates of 21.6% (29/134) were observed in sheep and of 11.4% (19/167) in goats. Furthermore, by qRT-PCR, HEV RNA was detected in four fecal samples collected from ovine farms. These findings seem to indicate that circulation of HEV among sheep and goat populations in Italy is more frequent than expected and it is not limited to a geographical area (southern Italy), considered at high-risk for human infection [76,77,78], where sustained viral circulation was demonstrated in pig herds or among wild boars [58,79,80]. In our analysis, many of the positive flocks were close to each other (Figure 2), remaining confined within an area of approximately 29.8 km^2^ in Piemonte and 14.9 km^2^ in Valle d’Aosta. Interestingly, the three sheep herds located a part in Valle d’Aosta, were found all seronegative for HEV. A similar geographical clustering was also observed for one roe deer and the two Alpine ibex possessing HEV antibodies (Figure 4), although no correlation in local distribution was revealed when comparing the seropositivities detected in wild ruminants with those from positive goat farms of the same region. These differences in HEV antibody detection rates could be related to a diverse distribution of HEV source within the territories investigated, underlining the needed of future studies enhancing the knowledges on HEV ecology in these specific areas. An age-related seroprevalence pattern was observed both in sheep and goats with the highest positivities in 3–4 years old animals and the lowest in the age groups <3 years. A similar age-related trend was observed in wild boars with the higher seroprevalence being reported in adults and subadults [12]. This trend may be consistent with a circulation of HEV among domestic ruminants at lower rates and more gradually. At the same time, the low seropositivity rates detected in older sheep and goats may be due to a fading immunity compatible with spillover HEV infection. Furthermore, the duration of protective immunity in domestic ruminants is still unknown and a short-term antibody response cannot be excluded. However, the low number of animals assessed in this study and the lack of homogeneity in the age stratification groups did not allow us to obtain any conclusive information. Additional studies based on a structured sampling could help to draw a more complete picture of HEV seroprevalence age-related pattern in sheep and goats.

In our molecular analysis, four sheep fecal specimens resulted positive for HEV RNA, although at low viral loads. Interestingly, all the molecularly positive sheep were seronegative, but sampled in both the farms in which HEV antibody detection rates were high (18.7% and 36.6%). This finding may be consistent with an active viral circulation within the two herds, at least at the time of sampling. In spite of several attempts to amplify viral RNA for sequencing by using broad-spectrum methods for species or family identification in various genome regions, none of the protocols were successful to generate sequence information.

Overall, the low prevalence rates and low viral loads observed in the virological investigation seem to suggest that sheep and goats are not a true reservoir of HEV, but more likely, they may be infected occasionally due to spillover events. It remains to establish exactly the HEV infection source for ruminants in the geographical setting investigated. Pigs and wild boars are known to be important reservoirs of HEV in industrialized countries. While data collected in northwestern Italy showed a low circulation of HEV in wild boars [70], local information on HEV epidemiology in pigs is still limited. With the exception of a survey performed in intensive pig farms from the Piemonte region in which viral RNA was found with a prevalence rate of 31.0% [81], similar studies in Valle d’Aosta, an area characterized by a high concentration of small free-range pig farms, are still lacking. In this contest, the close contact between infected pigs and domestic ruminants may represent a potential pathway of viral transmission. Otherwise, we cannot rule out that environmental contaminations favoured using swine manure in agriculture may constitute a source of indirect HEV exposure to animals not living in close contact but sharing the same pasture areas. In a study investigating viral contamination sources in the swine industry, more than half of the samples positive for HEV RNA was collected around farm buildings and slaughterhouse [82]. In addition, studying HEV prevalence in swine slurry from farms located in northern Italy resulted in 80% of positive samples [83].

Meanwhile, whether domestic ruminants may represent spillover or true reservoir of HEV needed to be further investigated by detailed observational studies testing sera and stools, as well as different tissue samples including liver, muscle, and spleen. Finally, animal experiments would be required to investigate this hypothesis.

## 5. Conclusions

This survey provides data on the presence and diffusion of HEV in wild and domestic ruminants in two Alpine Italian regions, Piemonte and Valle d’Aosta, where livestock animals are in close contact with wildlife animals, thus complicating HEV ecology. The high seroprevalence observed in sheep and goats should be studied more in depth in order to establish their role in HEV transmission. Information on the risk factors associated with HEV infection in small domestic ruminants is still sparse [52,55,57]. In a recent serosurvey in Portugal [55], HEV antibodies have been found in 16.6% of the sheep sera and in 29.3% of workers occupationally exposed to sheep or to ovine edible products, but only in 16.1% of the control population. More recently, in a molecular and serological study conducted in upper Egypt [57], Gt3 HEV RNA was identified either in goat milk and liver samples. Accordingly, devising surveillance plans to ascertain the viral hazards for humans associated with the consumption of products of ovine and caprine origins is strongly suggested.

## Figures and Tables

**Figure 1 animals-10-02351-f001:**
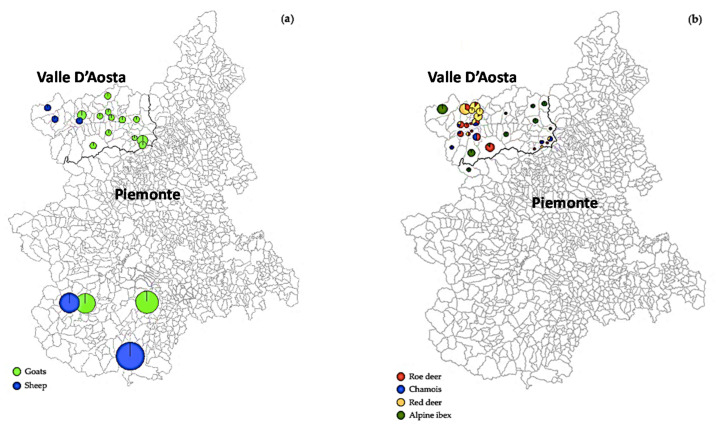
Mapping of sheep and goat farms (**a**) investigated and sites (**b**) in which wild ruminants were sampled.

**Figure 2 animals-10-02351-f002:**
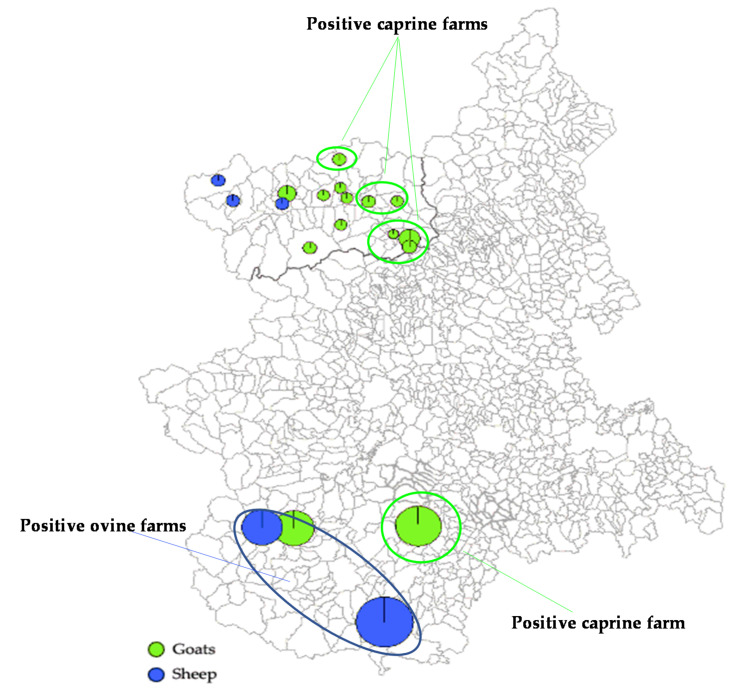
Geographical distribution (Piemonte and Valle d’Aosta) of sheep and goat farms positive for HEV antibodies.

**Figure 3 animals-10-02351-f003:**
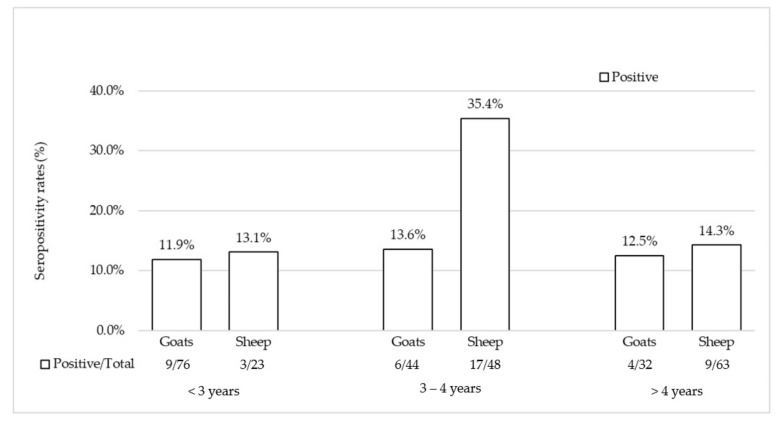
Antibodies to HEV in goat and sheep sera of different age groups.

**Figure 4 animals-10-02351-f004:**
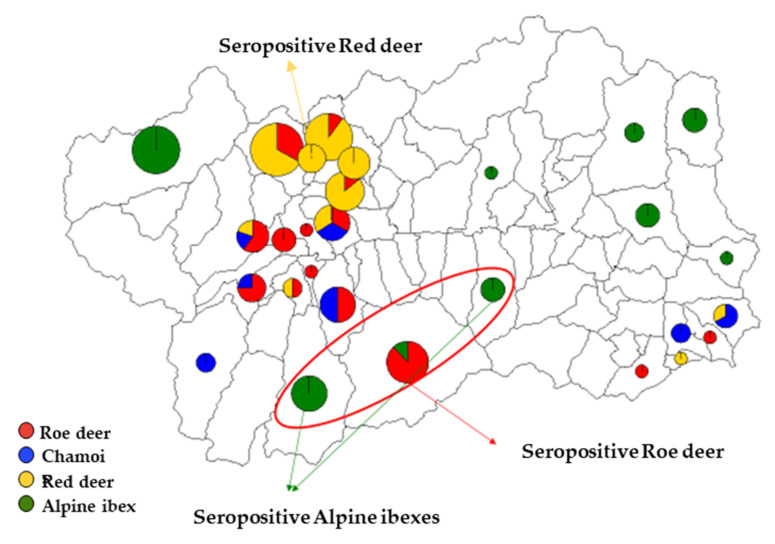
Maps of Valle d’Aosta Region showing the areas where seropositive wild ruminants were sampled.

**Table 1 animals-10-02351-t001:** Serological prevalence of Hepatitis E virus (HEV) in sheep and goat farms.

Species	Region	Locality	Positive/Total (%)	CI 95%
Sheep	Valle d’Aosta	Courmayeur	0/6 (0)	
Sheep	Valle d’Aosta	Morgex	0/6 (0)	
Sheep	Valle d’Aosta	Sarre	0/6 (0)	
Sheep	Piemonte	Sampeyre	15/41 (36.6)	22.1–53.1
Sheep	Piemonte	Roccaforte Mondovì	14/75 (18.7)	10.6–29.3
**Sheep—Total**			**29/134 (21.6)**	**15.0–29.6**
Goats	Valle d’Aosta	Verrayes	1/6 (16.7)	0.4–64.1
Goats	Valle d’Aosta	Perloz	3/6 (50.0)	11.8–88.2
Goats	Valle d’Aosta	Perloz	0/7 (0)	
Goats	Valle d’Aosta	Arnad	1/4 (25.0)	0.6–80.6
Goats	Valle d’Aosta	Pont St. Martin	1/7 (14.3)	0.4–57.9
Goats	Valle d’Aosta	Brusson	2/5 (40.0)	5.3–85.3
Goats	Valle d’Aosta	Chatillon	1/6 (16.7)	0.4–57.9
Goats	Valle d’Aosta	Gignod	0/5 (0)	
Goats	Valle d’Aosta	Gignod	0/5 (0)	
Goats	Valle d’Aosta	Cogne	0/6 (0)	
Goats	Valle d’Aosta	Nus	0/5 (0)	
Goats	Valle d’Aosta	Bionaz	0/5 (0)	
Goats	Valle d’Aosta	Quart	0/5 (0)	
Goats	Valle d’Aosta	Fenis	0/5 (0)	
Goats	Piemonte	Brossasco	10/40 (25.0)	13.0–41.2
Goats	Piemonte	Narzole	0/50 (0)	
**Goats—Total**			**19/167 (11.4)**	**7.0** **–** **17.2**
**Total**			**48/301 (15.9)**	**11.8** **–** **20.0**

**Table 2 animals-10-02351-t002:** Serological prevalence of HEV in wild ruminants.

Species	Region	Positive/Total (%)	CI 95%
Red deer	Valle d’Aosta	1/38 (2.6)	0.07–13.8
Roe deer	Valle d’Aosta	1/32 (3.1)	0.08–16.2
Alpine ibex	Valle d’Aosta	2/32 (6.3)	0.80–20.8
Chamois	Valle d’Aosta	0/13 (0)	
**Total**		**4/115 (3.5)**	**0.90–8.7**

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
