# Peer review of "Surveillance Study of Hepatitis E Virus (HEV) in Domestic and Wild Ruminants in Northwestern Italy"

_animals, 2020, doi:10.3390/ani10122351_

Round 1

Reviewer 1 Report

The manuscript of Palombieri and colleagues provides serological and molecular evidence of HEV infection in domesticated and wild ruminants within the same geographic localities.

They provide important additional data to the growing evidence of HEV infection in domesticated animals, which raises challenges for agriculture and human health in the future and definitely an important issue to be investigated in future studies as well.

The article describes the possible interference between wild and domesticated ruminants. I suggest to include the hypothetical transmission figure of HEV in ruminants, with key questions to be investigated in future studies. It would greatly improve the informative value of the article.

I miss some discussion ont he possible transmission routes between wild and domesticated ruminants – it may be provided via the suggested figure or included within the discussion part.

The detection of seropositivity in dedicated localities points toward future investigations in these territories to solve key questions. However low sampling numbers may also result in these differences. Why these regions are mostly affected, is it related to recent emergence of the virus or related to specific transmission patterns unique to these territories. These questions may be included in the discussion part as well, however it is a plain suggestion.

It is not surprising to fail verifying PCR and sequencing attempts with such a low viral copies number. The conclusion about the reservoir inability of these animals seems a good conclusion, it is higly supported by low viral load and low viral prevalence. However it stresses the question about the natural reservoir of the virus in these territories.

Overall I think the manuscript if acceptable in Animals Journal after minor revision, If the authors can include some more discussion about my remarks.

I would like to thank the authors for the great work and I am pleased to provide this review about this manuscript.

Author Response

Reviewer 1 (R1)

The manuscript of Palombieri and colleagues provides serological and molecular evidence of HEV infection in domesticated and wild ruminants within the same geographic localities.

They provide important additional data to the growing evidence of HEV infection in domesticated animals, which raises challenges for agriculture and human health in the future and definitely an important issue to be investigated in future studies as well.

R1.1 - The article describes the possible interference between wild and domesticated ruminants. I suggest to include the hypothetical transmission figure of HEV in ruminants, with key questions to be investigated in future studies. It would greatly improve the informative value of the article.

Reply to R1.1: in the revised manuscript the following section Previous molecular and serological survey conducted on conventional pig farms located in Piemonte Region reported rates of 31.0% and 50.2%, respectively [48], whilst data on HEV prevalence in swine in Valle d’Aosta are not available, although this geographical area is characterized by a high concentration of small free-range pig farms. Accordingly, we cannot rule out that environmental contaminations with HEV in the vicinities of herds may represent a potential source of infection for sheep and goats. In a study investigating viral contamination sources in the swine industry, more than half of the samples positive for HEV RNA has been collected around farm buildings and slaughterhouse [49].” was modified as follows (Lines 295-308 of the revised manuscript): It remains to establish exactly the HEV infection source for ruminants in the geographical setting investigated. Pigs and wild boars are known to be important reservoirs of HEV in industrialized countries. While data collected in Northwestern Italy showed a low circulation of HEV in wild boars [70], local information on HEV epidemiology in pigs is still limited. With the exception of a survey performed in intensive pig farms from Piemonte Region in which viral RNA has been found with a prevalence rate of 31.0% [81], similar studies in Valle d’Aosta, an area characterized by a high concentration of small free-range pig farms, are still lacking. In this contest, the close contact between infected pigs and domestic ruminants may represent a potential pathway of viral transmission. Otherwise, we cannot rule out that environmental contaminations favoured using swine manure in agriculture may constitute a source of indirect HEV exposure to animals not living in close contact but sharing the same pasture areas. In a study investigating viral contamination sources in the swine industry, more than half of the samples positive for HEV RNA has been collected around farm buildings and slaughterhouse [82]. Also, studying HEV prevalence in swine slurry from farms located in Northern Italy resulted in 80% of positive samples [83].”

R1.2 - I miss some discussion on the possible transmission routes between wild and domesticated ruminants – it may be provided via the suggested figure or included within the discussion part.

Reply to R1.2: the following sentence was added (Lines 270-273 of the revised manuscript): “A similar geographical clustering was also observed for one roe deer and the two Alpine ibex possessing HEV antibodies (Fig. 4), although none correlation in local distribution was revealed when comparing the seropositivities detected in wild ruminants with those from positive goat farms of the same Region.” (see also Reply to R.1.3).

R1.3 - The detection of seropositivity in dedicated localities points toward future investigations in these territories to solve key questions. However low sampling numbers may also result in these differences. Why these regions are mostly affected, is it related to recent emergence of the virus or related to specific transmission patterns unique to these territories. These questions may be included in the discussion part as well, however it is a plain suggestion.

Reply to R.1.3 – In the revised manuscript, the sentence “In our analysis, many of the positive flocks were close to each other (Fig. 2), remaining confined within an area of approximately 29.8 km2 in Piemonte and 14.9 km2 in Valle d’Aosta. This geographical clustering could explain the absence of seropositivity in three sheep herds located a part, in Valle d’Aosta.” was modified as follows (Lines 270-276 of the revised manuscript): “In our analysis, many of the positive flocks were close to each other (Fig. 2), remaining confined within an area of approximately 29.8 km2 in Piemonte and 14.9 km2 in Valle d’Aosta. Interestingly, the three sheep herds located a part in Valle d’Aosta were found all seronegative for HEV. A similar geographical clustering was also observed for one roe deer and the two Alpine ibex possessing HEV antibodies (Fig. 4), although none correlation in local distribution was revealed when comparing the seropositivities detected in wild ruminants with those from positive goat farms of the same Region. These differences in HEV antibody detection rates could be related to a diverse distribution of HEV source within the territories investigated, underlining the needed of future studies enhancing the knowledges on HEV ecology in these specific areas.”

It is not surprising to fail verifying PCR and sequencing attempts with such a low viral copies number. The conclusion about the reservoir inability of these animals seems a good conclusion, it is higly supported by low viral load and low viral prevalence. However it stresses the question about the natural reservoir of the virus in these territories.

Overall I think the manuscript if acceptable in Animals Journal after minor revision, If the authors can include some more discussion about my remarks.

I would like to thank the authors for the great work and I am pleased to provide this review about this manuscript.

Reviewer 2 Report

The authors carried out an interesting study of the seroprevalence of anti-HEV antibodies and the prevalence of viral RNA in wild and domestic ruminants in Italy. This work will enrich the literature on the epidemiology of the hepatitis E virus in industrialized countries and on possible secondary transmission routes for humans.

I think this article could be accepted with some minor modifications.

Line 48 : "nonenvelopped" ==> HEV was originally described as a non-enveloped virus but a quasi-enveloped form has been observed in blood (Takahashi et al. 2010 ; Chapuy-Regaud et al. 2017).

Table 1: Maybe you could add a final line at the end of the table "Overall" or "Total" to summarize all the samples tested (N), the number of positive samples and an overall seroprevalences in your study. Moreover, two intermediate lines with the total number of sheeps tested and then the same for goats could make the reading easier.

Figure 3 could be improved :

  • Indeed, I think it would be more understandable by removing the negative rates. Then we would see on the graph the three seroprevalences by agegroups for sheep and the three for goats.
  • The title of the axis should be added (Number of samples or Seropositivity rate (%). I also think there is an error on the chart. Indeed, if the y-axis shows the percentages, why is the 11.9% higher than the 13.6% in chart height? Or the y-axis indicates the number of samples (N positive and N negative) but this makes the reading more complex.
  • Three seroprevalences (positivity rates) of age groups per animal species would greatly simplify the reading of the graph while giving the same information. If you want to maintain the detail of thenumber of positives and negatives samples this can be shown in the table below the graph.

Line 197 to 204: I'm not sure I understood that part. Are the 4 positive sheep fecal samples found are false positive ? contaminations ?

Discussion part:

  • Calculated seroprevalences by age group in goats and sheeps should be discussed. Surprisingly, there is no cumulative effect of years on the observed seroprevalences ? or seroreversion in older animals ? Moreover, it seems surprising to see three almost identical seroprevalences in goats (11.9% 13.6% and 12.5%). Perhaps the sampling could have been improved if age was a variable of interest in the study?
  • Results regarding the prevalence of HEV RNA (4/134 or 0/134) in the samples should also be discussed.
  • A section on the limitations of the study should be added, in my opinion.

Author Response

Reviewer 2 (R2)

The authors carried out an interesting study of the seroprevalence of anti-HEV antibodies and the prevalence of viral RNA in wild and domestic ruminants in Italy. This work will enrich the literature on the epidemiology of the hepatitis E virus in industrialized countries and on possible secondary transmission routes for humans.

I think this article could be accepted with some minor modifications.

Specific comments:

R2.1 - Line 48: "nonenvelopped" ==> HEV was originally described as a non-enveloped virus but a quasi-enveloped form has been observed in blood (Takahashi et al. 2010; Chapuy-Regaud et al. 2017).

Reply to R2.1 – in agree with Referee, in the revised manuscript “non-enveloped” was replaced with “quasi-enveloped”. The references Takahashi et al. 2010 and Chapuy-Regaud et al. 2017 were added within the manuscript and in the references list.

Table 1:

R2.2 - Maybe you could add a final line at the end of the table "Overall" or "Total" to summarize all the samples tested (N), the number of positive samples and an overall seroprevalences in your study.

Reply to R2.2 – as suggested by the Referee, in the revised manuscript the Table 1 was modified by adding a final line in which the total of animals tested, and the positivity rate obtained, were reported.

R2.3 - Moreover, two intermediate lines with the total number of sheeps tested and then the same for goats could make the reading easier.

Reply to R2.3 – in the revised manuscript the Table 1 was modified as suggested by the Referee.

Figure 3 could be improved:

R2.4 - Indeed, I think it would be more understandable by removing the negative rates. Then we would see on the graph the three seroprevalences by agegroups for sheep and the three for goats.

Reply to R2.4 – in the revised manuscript the Figure 3 was modified as suggested by the Referee.

R2.5 - The title of the axis should be added (Number of samples or Seropositivity rate (%).

Reply to R2.5 – in the revised manuscript the title of the axis was modified as suggested by the Referee.

R2.6 - I also think there is an error on the chart. Indeed, if the y-axis shows the percentages, why is the 11.9% higher than the 13.6% in chart height? Or the y-axis indicates the number of samples (N positive and N negative) but this makes the reading more complex.

Reply to R2.6 – many thanks to the Referee. In the revised manuscript the chart was corrected.

R2.7 - Three seroprevalences (positivity rates) of age groups per animal species would greatly simplify the reading of the graph while giving the same information. If you want to maintain the detail of the number of positives and negatives samples this can be shown in the table below the graph.

Reply to R2.7 – in the graph only the positive rate for each age group per animal species was reported.

Discussion part:

R2.8 - Line 197 to 204: I'm not sure I understood that part. Are the 4 positive sheep fecal samples found are false positive ? contaminations ?

Reply to R2.8 – many thanks to the Referee. In order to explain better the results obtained, in the revised manuscript the sentence “HEV RNA was detected by qRT-PCR only in sheep faecal specimens with a prevalence rate of 3.0% (4/134)” was modified as follows (Lines 221-223 of the revised manuscript): “HEV RNA was detected by qRT-PCR in four sheep faecal specimens (3.0%; 4/134) collected from the two herds in which HEV antibody detection rates were 18.7% and 36.6%, respectively.” Furthermore, the sentence: None of the sheep positive to viral RNA possessed HEV antibodies even if all the animals were living in two serologically positive herds located in Piemonte” was removed.

R2.9 - Calculated seroprevalences by age group in goats and sheeps should be discussed. Surprisingly, there is no cumulative effect of years on the observed seroprevalences ? or seroreversion in older animals ?

Reply to R2.9 – In the revised manuscript the following sentence was added (Lines 279-283 of the revised manuscript): “This trend may be consistent with a circulation of HEV among domestic ruminants at lower rates and more gradually. At the same time, the low seropositivity rates detected in older sheep and goats may be due to a fading immunity, compatible with spillover HEV infection. Furthermore, the duration of protective immunity in domestic ruminants is still unknown and a short-term antibody response cannot be excluded”.

R2.10 - Moreover, it seems surprising to see three almost identical seroprevalences in goats (11.9% 13.6% and 12.5%). Perhaps the sampling could have been improved if age was a variable of interest in the study?

Reply to R2.10 – In agree with the Referee the sampling could have been improved in order to obtain more precise information on the association between HEV seropositivity and age. However, in our analysis, the samples collection was performed using a convenience sampling strategy. Accordingly, a more homogenous age stratification for each domestic ruminant species sampled in this study was not obtained.

R2.11 - Results regarding the prevalence of HEV RNA (4/134 or 0/134) in the samples should also be discussed.

Reply to R2.11 – In the revised manuscript (section discussion), the following sentence: “In our molecular analysis, only four fecal specimens collected from seronegative sheep resulted positive at low viral loads.” was modified as follows (Lines 288-291 of the revised manuscript): In our molecular analysis, four sheep fecal specimens resulted positive for HEV RNA, although at low viral loads. Interestingly, all the molecularly positive sheep were seronegative, but sampled in both the farms in which HEV antibody detection rates were high (18.7% and 36.6%). This finding may be consistent with an active viral circulation within the two herds, at least at the time of sampling.

R2.12 - A section on the limitations of the study should be added, in my opinion.

Reply to R2.12 – In the revised manuscript the following sentence was added (Lines 283-287 of the revised manuscript): “However, the low number of animals assessed in this study and the lack of homogeneity in the age stratification groups did not allow us to obtain any conclusive information. Additional studies based on a structured sampling could help to draw a more complete picture of HEV seroprevalence age-related pattern in sheep and goats.”.

Reviewer 3 Report

Reviewer Comments to Authors

(1.) Introduction:

Please write in this section more information on hepatitis E virus infection in animals from other Southern European countries. In this regard, please write the following scientific publications (references) in this section (and in the section "References"):

  • Zele D, Barry AF, Hakze-van der Honing RW, Vengust G, van der Poel WH. Prevalence of anti-Hepatitis E virus antibodies and first detection of hepatitis E virus in wild boar in Slovenia. Vector Borne Zoonotic Dis 2016; 16(1): 71-74. [DOI: 10.1089/vbz.2015.1819] [PMID: 26757050]
  • Tsachev I, Baymakova M, Pepovich R, Palova N, Marutsov P, Gospodinova K, Kundurzhiev T, Ciccozzi M. High seroprevalence of hepatitis E virus infection among East Balkan swine (Sus scrofa) in Bulgaria: preliminary results. Pathogens 2020; 9(11): 911. [DOI: 10.3390/pathogens9110911] [PMID: 33153218]
  • Lupulovic D, Lazic S, Prodanov-Radulovic J, Jimenez de Oya N, Escribano-Romero E, Saiz JC, Petrovic T. First serological study of hepatitis E virus infection in backyard pigs from Serbia. Food Environ Virol 2010; 2: 110-113. [DOI: 10.1007/s12560-010-9033-6]
  • Tsachev I, Baymakova M, Ciccozzi M, Pepovich R, Kundurzhiev T, Marutsov P, Dimitrov KK, Gospodinova K, Pishmisheva M, Pekova L. Seroprevalence of hepatitis E virus infection in pigs from Southern Bulgaria. Vector Borne Zoonotic Dis 2019; 19(10): 767-772. [DOI: 10.1089/vbz.2018.2430] [PMID: 31017536]
  • Jemersic L, Keros T, Maltar L, Barbic L, Vilibic-Cavlek T, Jelicic P, Rode OD, Prpic J. Differences in hepatitis E virus (HEV) presence in naturally infected seropositive domestic pigs and wild boars - an indication of wild boars having an important role in HEV epidemiology. Vet Arhiv 2017; 87(6): 651-663. [DOI: 10.24099/vet.arhiv.170208]
  • Siochu A, Tzika E, Alexopoulos C, Kyriakis SC, Froesner G. First report of serological evidence of hepatitis E virus infection in swine in Northern Greece. Acta Vet 2009; 59(2-3): 205-211. [DOI: 10.2298/AVB0903205S]
  • Boadella M, Ruiz-Fons JF, Vicente J, Martin M, Segales J, Gortazar C. Seroprevalence evolution of selected pathogens in Iberian wild boar. Transbound Emerg Dis 2012; 59(5): 395-404. [DOI: 10.1111/j.1865-1682.2011.01285.x] [PMID: 22168900]
  • de Deus N, Peralta B, Pina S, Allepuz A, Mateu E, Vidal D, et al. Epidemiological study of hepatitis E virus infection in European wild boars (Sus scrofa) in Spain. Vet Microbiol 2008; 129(1-2): 163-170. [DOI: 10.1016/j.vetmic.2007.11.002] [PMID: 18082340]

(2.) Materials & Methods:

  • Please write the exact period for "regular hunting season" (dates, from ... to ...) (Line 100).
  • Please write "Sensitivity and Specificity" for your use ELISA test (Lines 112-119).
  • Please write the subsection "Ethics Statement" (after subsection "Statistical Analysis").

(3.) Results:
I have no recommendations for this section.

(4.) Discussion:
I have no recommendations for this section.

(5.) References:
Please see the recommendations and comments on the section "Introduction".

(6.) Tables:
I have no recommendations for this section.

(7.) Figures:
I have no recommendations for this section.

Author Response

Reviewer 3 (R3)

R3.1 - Please write in this section more information on hepatitis E virus infection in animals from other Southern European countries. In this regard, please write the following scientific publications (references) in this section (and in the section "References"):

Zele D, Barry AF, Hakze-van der Honing RW, Vengust G, van der Poel WH. Prevalence of anti-Hepatitis E virus antibodies and first detection of hepatitis E virus in wild boar in Slovenia. Vector Borne Zoonotic Dis 2016; 16(1): 71-74. [DOI: 10.1089/vbz.2015.1819] [PMID: 26757050]

Tsachev I, Baymakova M, Pepovich R, Palova N, Marutsov P, Gospodinova K, Kundurzhiev T, Ciccozzi M. High seroprevalence of hepatitis E virus infection among East Balkan swine (Sus scrofa) in Bulgaria: preliminary results. Pathogens 2020; 9(11): 911. [DOI: 10.3390/pathogens9110911] [PMID: 33153218]

Lupulovic D, Lazic S, Prodanov-Radulovic J, Jimenez de Oya N, Escribano-Romero E, Saiz JC, Petrovic T. First serological study of hepatitis E virus infection in backyard pigs from Serbia. Food Environ Virol 2010; 2: 110-113. [DOI: 10.1007/s12560-010-9033-6]

Tsachev I, Baymakova M, Ciccozzi M, Pepovich R, Kundurzhiev T, Marutsov P, Dimitrov KK, Gospodinova K, Pishmisheva M, Pekova L. Seroprevalence of hepatitis E virus infection in pigs from Southern Bulgaria. Vector Borne Zoonotic Dis 2019; 19(10): 767-772. [DOI: 10.1089/vbz.2018.2430] [PMID: 31017536]

Jemersic L, Keros T, Maltar L, Barbic L, Vilibic-Cavlek T, Jelicic P, Rode OD, Prpic J. Differences in hepatitis E virus (HEV) presence in naturally infected seropositive domestic pigs and wild boars - an indication of wild boars having an important role in HEV epidemiology. Vet Arhiv 2017; 87(6): 651-663. [DOI: 10.24099/vet.arhiv.170208]

Siochu A, Tzika E, Alexopoulos C, Kyriakis SC, Froesner G. First report of serological evidence of hepatitis E virus infection in swine in Northern Greece. Acta Vet 2009; 59(2-3): 205-211. [DOI: 10.2298/AVB0903205S]

Boadella M, Ruiz-Fons JF, Vicente J, Martin M, Segales J, Gortazar C. Seroprevalence evolution of selected pathogens in Iberian wild boar. Transbound Emerg Dis 2012; 59(5): 395-404. [DOI: 10.1111/j.1865-1682.2011.01285.x] [PMID: 22168900]

de Deus N, Peralta B, Pina S, Allepuz A, Mateu E, Vidal D, et al. Epidemiological study of hepatitis E virus infection in European wild boars (Sus scrofa) in Spain. Vet Microbiol 2008; 129(1-2): 163-170. [DOI: 10.1016/j.vetmic.2007.11.002] [PMID: 18082340]

Reply to R3.1 – As suggested by Referee additional information on HEV infection was reported in the revised manuscript. Furthermore, the References indicated was included in the manuscript. In the introduction section, the following sentence was added (Lines 62-68 of the revised manuscript):“Since the first identification of Gt3 HEV in swine [13], several molecular and serological surveys showed high prevalence in pigs and wild boars worldwide[12]. In Europe, investigations performed in swine herds revealed seroprevalences estimated between 27% and 100% [14-23] with molecular detection rates of 0.9% to 87.5% [24-33]. Similarly, epidemiological studies performed in wild boar populations reported antibody detection rates ranging from 12.5% to 57.4% and molecular prevalence of 0.3% to 68.2% [12,19,34-38].”

(2.) Materials & Methods:

R3.2 - Please write the exact period for "regular hunting season" (dates, from ... to ...) (Line 100).

Reply to R3.2 – as suggested by Referee, in the revised manuscript the exact period of the regular hunting season was indicated as follows (Line 108 of the revised manuscript): “The regular hunting season (from September 9 to December 18)”.

R3.3 - Please write "Sensitivity and Specificity" for your use ELISA test (Lines 112-119).

Reply to R3.3 – In the revised manuscript the following sentence was added (Lines 127-133 of the revised manuscript): “When Comparing with other HEV antibody detection ELISAs, the assay employed in this study was found to have a high sensitivity, although it seems to possess a lower specificity (Krumbholz et al., 2013). This antigen, widely used to detect Gt1-Gt4 HEV antibodies in humans, has also been successfully used to evaluate HEV seroconversion in animals infected either under natural or experimental conditions with novel members of the species Orthohepevirus A, such as Gt7 and Gt8 (Bassal et al., 2019; Wang et al., 2019), as well as to detect specific IgG antibodies in ferrets and foxes that tested positive, on molecular investigations, for Orthohepevirus C RNA (Eiden et al., 2020; Raj et al., 2012).

R3.4 - Please write the subsection "Ethics Statement" (after subsection "Statistical Analysis").

Reply to R3.4 – as suggested by Referee, in the revised manuscript (section Material and methods), the subsection "Ethics Statement" was added as follows (Lines 165-169 of the revised manuscript): “For this study no ethical approval was required (Decreto Legislativo 4 March 2014, n. 26). Serum and faecal samples from wild ruminants were collected during the routinely monitoring activities performed by the National Reference Centre for Wild Animal Diseases (CeRMAS – IZS PLV), whilst samples from sheep and goats were collected only for veterinary diagnostic purposes.”

(3.) Results:

I have no recommendations for this section.

(4.) Discussion:

I have no recommendations for this section.

R3.5 - (5.) References:

Please see the recommendations and comments on the section "Introduction".

(6.) Tables:

I have no recommendations for this section.

(7.) Figures:

I have no recommendations for this section.

Round 2

Reviewer 3 Report

Reviewer Comments to Authors

I read carefully the revised manuscript. I'm happy with that which I saw and read. I have no recommendations or comments to the revised manuscript.

My recommendation: accept the manuscript for publication in this form.